# Understanding Insulin Actions Beyond Glycemic Control: A Narrative Review

**DOI:** 10.3390/jcm14145039

**Published:** 2025-07-16

**Authors:** Rayan Riachi, Elie Khalife, Andrzej Kędzia, Elżbieta Niechciał

**Affiliations:** Department of Pediatric Diabetes, Clinical Auxology and Obesity, Poznan University of Medical Sciences, 60-572 Poznan, Poland; rayanalriachi@hotmail.com (R.R.); elie.r.khalife01@gmail.com (E.K.); akedzia@ump.edu.pl (A.K.)

**Keywords:** insulin, insulin action beyond glycemia control, protein metabolism, lipid metabolism, skeletal muscle remodeling, insulin resistance, central nervous system, cognitive function, neurodegeneration, reproductive system

## Abstract

Insulin, traditionally recognized for its pivotal role in glycemic regulation, exerts extensive effects beyond glucose homeostasis, influencing multiple physiological systems. This narrative review explores the multifaceted actions of insulin, emphasizing its impact on skeletal muscle remodeling, protein and lipid metabolism, growth, reproductive health, and the central nervous system. **Methods**: An in-depth review of articles with evidence-based research discussing insulin actions beyond glycemic control was conducted in this review paper. **Results**: Insulin directly influences lipid and protein metabolism as well as growth hormone levels. This hormone provides a protective effect on the skeletal and central nervous systems, helping to maintain homeostasis and potentially reducing the risk of certain disorders such as Alzheimer’s disease. The significance of insulin balance in the reproductive system is also crucial, with recent research indicating that insulin plays a role in worsening symptoms and complications associated with polycystic ovary syndrome. This review underscores the importance of maintaining proper insulin levels to lower the risk of insulin resistance. Ongoing research aims to deepen our understanding of insulin’s functions, which are essential for preventing specific diseases and developing new treatment strategies. **Conclusions**: Insulin’s action extends far beyond glucose metabolism, affecting many systems and preventing pathological changes in some.

## 1. Introduction

Insulin, a polypeptide hormone predominantly secreted by the β-cells of the islets of Langerhans in the pancreas, plays a central role in regulating blood glucose levels, acting in coordination with glucagon. Insulin facilitates anabolic pathways, whereas glucagon governs catabolic processes [1]. In its mature form, insulin comprises two polypeptide chains: a 30-residue B chain and a 21-residue A chain, connected by two disulfide bonds (CysB7 to CysA7 and CysB19 to CysA21), with an additional disulfide bond within the A chain (CysA6 to CysA11). This mature hormone is derived from “proinsulin,” a precursor molecule consisting of a single polypeptide chain, where the C terminus of the B chain is linked to the N terminus of the A chain by a structure known as the “C-peptide” [2].

Insulin is a key hormone in glucose metabolism, but its effects extend beyond simply lowering blood glucose levels. This pleiotropic hormone decreases blood glucose levels by affecting many physiological pathways. Insulin increases glucose uptake at the level of many organs via the glucose transporter type 4 (GLUT-4), notably in the liver, skeletal muscles, and adipose tissue, which is essential in regulating glycemia [3,4]. Glucagon secretion is directly affected by insulin levels, working as a feedback mechanism that keeps blood glucose levels balanced. Insulin directly inhibits glucagon secretion during the fed state, pushing the body to use glucose as the primary energy source for ATP production [5]. While the direct action of insulin on hepatic cellular pathways is known, particularly the activation of glycolysis, glycogenesis, and lipogenesis, new studies are targeting the indirect effects of insulin on the liver. These indirect effects include the decrease in glycerol and free fatty acids released from adipose tissues and the suppression of hepatic gluconeogenic genes (phosphoenolpyruvate carboxykinase (PEPCK) and glucose-6-phosphatase (G6Pase) [6]. Lastly, the action of insulin on the central nervous system is becoming a subject of interest for many [7,8]. Although the results of trials can be conflicting [9], some studies have shown that hypothalamic insulin action decreases hepatic glucose production and can be a target for new drugs and treatments for diabetes [10].

Insulin is mainly used to manage type 1 diabetes (T1D). The discovery of insulin in 1921 by Sir Frederick G. Banting was unquestionably one of the top scientific achievements of the 20th century. Since then, insulin has saved the lives of countless individuals suffering from T1D. However, it might also be prescribed to patients with type 2 diabetes (T2D), pregnant women developing gestational diabetes, and some people with other types of diabetes, or in certain situations to manage hyperglycemia [11].

The physiological effects of insulin on carbohydrate metabolism are well recognized. Less understood but far more interesting are the effects of this hormone that go beyond glycemic control to help sense, integrate, and maintain energy balance. In addition to its metabolic effects, insulin stimulates cell growth and differentiation, including pancreatic β-cell growth and survival [12]. Recently, there has been more attention to the significant impact of insulin on the reproductive system, particularly on women. An enormous number of studies have shown that there is a link between insulin excess and hyperandrogenism, polycystic ovary syndrome (PCOS), ovarian follicle dysfunction, and endometrial hyperplasia [13]. Moreover, research in recent years has revealed several important connections between insulin and cognitive processes, particularly in the context of aging, metabolic disorders, and neurodegenerative diseases [14]. This review provides a comprehensive overview of the effects of insulin actions beyond regulating glucose metabolism.

## 2. Materials and Methods

This narrative review investigates evidence on the extrapancreatic effects of insulin. The paper aims to identify the most clinically relevant and current literature, thoroughly exploring insulin effects beyond glycemic control. The analysis was not conducted through a systematic literature review. Clear and comprehensive inclusion and exclusion criteria were established before starting the review process to mitigate selection bias. The search was carried out across multiple databases to capture a wide range of relevant studies, reducing the risk of overlooking critical data. The screening process was performed by each author independently to assess the most significant evidence available and to minimize subjective judgment and errors. Finally, the entire selection process was documented in a flow diagram (Figure 1).

The inclusion criteria were original research articles, including observational studies (cohort, case-control, cross-sectional), clinical trials, reviews, systematic reviews, and meta-analyses relevant to the extrapancreatic effects of insulin. The exclusion criteria included non-English language papers, studies with insufficient data, case reports, editorials, commentaries, non-peer-reviewed articles, duplicates, unavailable full texts, or abstract-only papers. The following electronic databases were searched for the most important full-text articles in English: PubMed, Google Scholar, EMBASE, Scopus, National Center for Biotechnology Information (NCBI), and Web of Science. All the articles published up to 2025 were reviewed for relevance to the research question, abstract, and full text.

The search was carried out using the following keywords: insulin, action, physiology, protein metabolism, lipid metabolism, skeletal muscle remodeling, insulin resistance, insulin-like growth factor-1, growth hormone, central nervous system, cognitive function, Alzheimer’s, neurodegeneration, polycystic ovary syndrome, reproductive system, cardiovascular disease, nitric oxide, endothelium, heart, myocardium, kidney, hypertension, and renal glucogenesis.

The search was conducted between December 2024 and July 2025. The work provides a theoretical point of view of the effects of insulin beyond glycemic control.

## 3. Results

Traditionally recognized for its pivotal role in glycemic regulation, insulin exerts extensive effects beyond glucose homeostasis, influencing multiple physiological systems. Figure 2 gives a general overview of insulin’s actions outside glucose regulation.

### 3.1. Effects of Insulin on Skeletal Remodeling

Over the past few decades, the skeleton has been redefined as a hormonally active organ that secretes osteokines, which play a crucial role in regulating energy metabolism and mineral homeostasis [15]. The discovery of bone-derived proteins that enter the bloodstream and influence systemic homeostasis has transformed the understanding of skeletal physiology [16]. Among these, osteocalcin has been shown to stimulate insulin production in pancreatic β-cells, enhance adiponectin secretion from adipose tissue, and improve glucose uptake in adipocytes. This interplay between bone and metabolic regulation highlights the essential role of insulin not only in glucose metabolism but also in skeletal function, where it regulates osteocalcin production by controlling its gene transcription and influencing its posttranslational modification [17].

Beyond its metabolic effects, insulin is a key regulator of osteoblast function, promoting collagen synthesis and the production of alkaline phosphatase (ALP), both essential for bone formation [18,19,20]. In a preclinical study conducted by Studenstowa et al., mice were used to assess the importance of insulin in bone homeostasis. The insulin receptor gene was deleted from the genome of S100a4 cells, which are important cells that regulate calcium binding, and bone samples were extracted at weeks 15 and 48. While no change in bone formation was detected at 15 weeks, mutated mice showed a substantial decrease in trabecular bone volume, bone volume fraction, and torsional rigidity, compared to wild-type mice at week 48. This study proved a direct influence of insulin on bone metabolism, with its strongest association during aging and maintenance of bone [18].

Insulin influences osteogenesis through both direct and indirect mechanisms. Directly, it acts via insulin and insulin-like growth factor-1 (IGF-1) receptors on osteoblasts, while indirectly, it regulates blood glucose levels and modulates the activity of parathyroid hormone, IGF-1, and vitamin D. Disruptions in insulin signaling can negatively impact bone health, increasing the risk of osteoporosis and fractures. For example, individuals with insulin resistance, T2D, or T1D may experience alterations in bone metabolism, including bone mineral density (BMD). Particularly, T1D due to insulin deficiency impacts bone health by decreasing BMD. Since T1D usually begins in adolescence—a critical time for rapid bone growth—affected individuals often develop lower BMD, which may increase their risk of bone weakness and fractures later in life. A recent meta-analysis of 14 observational studies demonstrated 2066 fracture events among 27,300 adults with T1D (7.6%) and 136,579 fracture events in 4,364,125 individuals without diabetes (3.1%) [21]. In contrast, those with T2D generally exhibit higher BMD, which may be attributed to an increased body mass index (BMI) [22]. Although individuals with T2D often have BMD, they still face a greater risk of fractures compared to the general population. This paradox highlights the complex changes in diabetic bone health, which impair bone quality beyond what is measurable through BMD alone. However, this impact might depend on the diabetes treatment method. The longitudinal study, which comprised 110 women with diabetes and examined the influence of insulin initiation treatment on BMD, showed that individuals using insulin had a greater loss of BMD at the femoral neck than women who did not use insulin [22].

Within an optimal concentration range, insulin enhances bone synthesis markers, including glucose uptake, ALP production, and collagen synthesis. Osteoblasts lacking insulin receptors (IR) exhibit impaired osteogenic differentiation, reduced ALP levels, and decreased expression of runt-related transcription factor 2 (Runx2) [23]. Insulin further promotes osteoblast differentiation by suppressing Twist2, a known inhibitor of Runx2 [17]. Endogenous insulin promotes the liver’s expression of growth hormone, leading to IGF-1 production. Thus, in the case of insulin deficiency in T1D, this would result in reduced osteoblast activity and diminished bone formation [24]. Osteoblastogenesis is hampered when the Wnt/β-catenin pathway is inhibited. This pathway is suppressed by a lack of insulin or IGF-1 signaling. Insulin resistance and decreased insulin or IGF-1 levels affect Akt phosphorylation and phosphorylated GSK3β levels, which promote the breakdown of β-catenin. Therefore, GSK3β may act as a molecular link between insulin signaling and the Wnt/β-catenin pathway in controlling bone and insulin balance. The Wnt/β-catenin pathway is further suppressed by Wnt inhibitors, which are more highly expressed in the absence of insulin. Ultimately, this reduces osteoblast differentiation, limiting the bone’s capacity to resist resorption [25]. Low bone turnover is found in patients with T1D, which makes their bones more brittle. Studies have demonstrated that T1D patients have lower bone resorption markers, such as C-terminal cross-linking telopeptide (CTX), and bone formation markers, such as osteocalcin (OCN), than healthy people. These patients frequently exhibit decreased BMD from an early age, most likely due to a total lack of endogenous insulin and the incapacity of administered insulin to mimic its natural secretion patterns [26]. Insulin-like growth factor binding proteins (IGFBPs) act to increase IGF-1 half-life and its ability to be transported to IGF-1 receptors, which would result in growth and proliferation. However, IGFBP-1 inhibits this effect by reducing the affinity of bound IGF-1 for IGF receptors. Insulin is a key negative regulator of IGFBP-1, thus increasing IGF-1 action [24,27]. A cross-sectional study was carried out at the National Institutes of Health, involving a comparison of individuals with congenital generalized lipodystrophy (CGL, *n* = 23), complete insulin receptor deficiency (INSR−/−, *n* = 13), partial deficiency (INSR+/−, *n* = 17), and type B insulin resistance (TBIR, *n* = 8). The primary outcomes measured included standardized scores (SDS) for height, BMI, BMD, and levels of insulin, IGF-1, and IGF binding proteins (IGFBP-1 and -3). Patients with INSR−/− demonstrated significantly elevated insulin and IGFBP-1 levels but considerably lower BMI, height, BMD, and IGFBP-3 than those with CGL. Individuals with TBIR experienced normalization of hormonal parameters upon remission. These results indicate that insulin receptor signaling fosters growth and bone density through direct and indirect pathways involving the GH–IGF-1 axis [27].

IR is present in osteoblasts and osteoclasts, while insulin receptor substrates (IRS) are exclusive to osteoblasts. IRS plays a critical role in insulin and IGF-1 signaling by regulating osteoblast proliferation, survival, and energy metabolism. Additionally, it influences the production of receptor activators of nuclear factor-kappa B ligand (RANKL), a key factor in osteoclast differentiation [23]. Impaired IR function, as seen in hyperinsulinemia, has been associated with reduced growth and lower BMD, whereas excessive insulin receptor signaling, as observed in congenital generalized lipodystrophy (CGL), is linked to accelerated growth and higher BMD [27].

### 3.2. The Interplay Between Insulin, Insulin-like Growth Factor-1, and Growth Hormone

Insulin and IGF-1 are two of the most significant hormones that play critical roles in growth, metabolism, and cellular function. However, insulin binds with a high affinity to IR, while IGF-1 binds with the highest affinity to the IGF-1 receptor (IGF1R); both signaling pathways are closely related. Insulin, as well as IGF-1, activate similar intracellular signaling pathways, particularly the phosphoinositide 3-kinase (PI3K)/Akt pathway and the mitogen-activated protein kinase (MAPK) pathway, which, in turn, regulate gene transcription, glucose, lipid, and protein metabolism as well as cell growth and differentiation. IGF-1 shares a molecular structure similar to insulin and can activate downstream signaling pathways through cross-receptor interactions. It also binds to IR and specifically activates IRS [23]. Additionally, insulin acts as a key negative regulator of IGFBP-1, which inhibits growth by reducing the affinity of IGF-1 for its receptors [27].

### 3.3. The Link Between Insulin and Protein and Lipid Metabolism

Studies indicate that insulin’s main anabolic effect on protein metabolism at the whole-body level is the suppression of whole-body proteolysis. Felig et al. conducted seminal metabolic studies identifying major biochemical abnormalities in nitrogen-containing compounds, particularly branched-chain amino acids (BCAAs), which were significantly elevated in insulin-deficient diabetes, especially in ketoacidosis [28,29]. Another marker of muscle protein degradation, urinary 3-methylhistidine excretion, was increased by ≥40% in individuals with T1D despite insulin therapy, indicating incomplete suppression of protein catabolism [30].

Insulin administration led to a dose-dependent reduction in plasma [24,27,31,32], and intracellular amino acid concentrations [15], particularly BCAAs, while also enhancing amino acid transport [16], mainly through individuals A and X–C systems [33]. Insulin further influences muscle metabolism by increasing blood flow, either systemically or through localized infusion [34,35,36], largely via nitric oxide production [37]. This improved circulation facilitates the delivery of anabolic substrates such as amino acids and glucose, thereby promoting protein synthesis. Additionally, insulin affects amino acid kinetics in the splanchnic region, where, in post-absorptive healthy individuals, protein synthesis exceeds protein degradation [38].

A deeper understanding of insulin’s role in proteostasis is crucial for both normal physiology and pathological conditions like T1D, where insulin deficiency disrupts protein metabolism [39]. Mechanistically, insulin binds to its receptor, activating the PI3K-Akt-mTORC1 pathway, which regulates metabolism and muscle growth [40]. Akt activation mediates insulin’s effects on muscle, enhancing glucose uptake, muscle growth, and protein turnover [41], while also influencing protein metabolism in conditions such as sepsis.

At the molecular level, insulin promotes protein synthesis by accelerating mRNA translation, stimulating initiation, and increasing eIF4E bioavailability through 4E-BP1. It also activates S6K1, which phosphorylates ribosomal protein S6 to enhance translation further [41]. Surprisingly, a screen of insulin receptor-associated proteins identified RNA Polymerase II (Pol II) as a major binding partner. This finding suggests insulin may directly influence gene expression by promoting insulin receptor translocation to chromatin, where it binds gene promoters with high specificity [42]. Ultimately, insulin significantly enhances net protein balance and acquisition, reinforcing its critical role in metabolism and growth [43].

Insulin is a critical regulator of glucose and lipid metabolism, which plays a central role in maintaining metabolic homeostasis. Notably, free fatty acids (FFAs) exhibit a response to insulin that is approximately three times faster than that of glucose, emphasizing insulin’s predominant influence on lipid metabolism [44]. This difference in response underscores the intricate nature of insulin’s metabolic control, particularly in insulin resistance.

The role of insulin in amino acid metabolism can be supported clinically by the studied relation between T2D and sarcopenia. Sarcopenia, which is defined as a progressive skeletal muscle disorder that leads to muscle atrophy and weakness [45], is strongly associated with T2D [46,47]. While the exact pathophysiology is not well understood, it is hypothesized that insulin resistance is one of the major causes [48]. As mentioned previously, insulin plays a major role in amino acid uptake and protein synthesis in skeletal muscle cells. Synthesis is primarily mediated by the IGF-1-PI3K-Akt-mTOR pathway, activated by insulin. Protein degradation is mainly mediated by an ATP-dependent ubiquitin proteasome pathway (UPP), which is downregulated by insulin [49]. In the case of IR, the imbalance between catabolic and anabolic pathways, along with other factors like inflammation and oxidative stress, is a factor contributing to the development of sarcopenia. Based on this, the benefits of using anti-diabetic medication and insulin in the context of sarcopenia are being studied [50]. Recently, Liu L et al. found that using insulin glargine, a long-acting insulin, effectively prevented muscle loss in elderly T2D patients [51]. This study aimed to examine the effects of insulin glargine and exenatide (glucagon-like peptide-1 receptor agonists) on the muscle mass of patients with newly diagnosed T2D. The study comprised 76 individuals aged 18−70, with a BMI over 24 kg/m2 and HbA1c levels between 7% and 10%. Participants were randomly assigned 1:1 to either the insulin glargine or the exenatide group. The cross-sectional Dixon-fat MRI assessed the psoas muscle area (PMA) changes at the fourth lumbar vertebra. At the baseline, this study did not find notable differences in PMA between the insulin glargine and exenatide groups. Following treatment, PMA showed a tendency to increase by 13.13 mm^2^ (range: –215.52 to 280.80 mm^2^) in the insulin glargine group and decrease by 149.09 mm^2^ (range: 322.90 to –56.39 mm^2^) in the exenatide group, with both changes not reaching statistical significance (*p* > 0.05). Subgroup analysis indicated that, in patients with a BMI less than 28 kg/m^2^, PMA rose by 560.64 mm^2^ (range: 77.88 to 1043.40 mm^2^) in the insulin group compared to the exenatide group after adjusting for gender and age (*p* = 0.031). Interaction analysis revealed a significant interaction between BMI and treatment (*p* = 0.009). However, no significant interactions were observed among subgroups with a BMI of 28 kg/m^2^ or higher, or across different genders and age groups. Therefore, this study found that insulin glargine may relatively increase PMA in individuals with T2D having a BMI less than 28 kg/m^2^ compared to exenatide [51].

One of insulin’s key metabolic effects is its ability to suppress the systemic flux of nonesterified fatty acids (NEFAs) more effectively than glucose or other metabolites. This suppression primarily occurs through inhibiting intracellular triglyceride (TG) breakdown within adipose tissue. By limiting lipolysis, insulin plays an essential role in preventing diabetic ketoacidosis while also mitigating the harmful lipotoxic effects of excess NEFAs on lean tissues, which contribute to diabetes-related complications. Additionally, insulin regulates several lipid metabolic processes in adipose tissue, including fatty acid esterification, glycerol and TG synthesis, lipogenesis, and potentially fatty acid oxidation, thereby promoting the efficient sequestration of dietary fatty acids in the postprandial state [52].

In addition, insulin exerts its effects through a complex signaling cascade involving PI3K and AKT, which regulate both hepatic and systemic metabolic homeostasis. Beyond its role in adipose tissue, insulin is a crucial modulator of hepatic metabolism, directing the storage and distribution of nutrients. Under postprandial conditions, insulin stimulates the liver to convert excess nutrients into triglycerides, cholesterol, and glycogen. This process is mediated through modifications in gene expression and posttranslational mechanisms that enhance lipogenesis. However, in insulin-resistant states such as T2D, hepatic insulin signaling remains active in driving lipid synthesis while failing to suppress glucose production adequately. Furthermore, insulin plays a role in regulating hepatic fatty acid oxidation by modulating mitochondrial enzyme activity, thereby influencing overall lipid balance. The disruption of these metabolic processes contributes to hypertriglyceridemia and hyperglycemia, exacerbating metabolic imbalances and elevating the risk of cardiovascular complications [53].

### 3.4. Effect of Insulin on the Central Nervous System

Glucose is the primary energy source of the brain. It crosses the blood–brain barrier (BBB) via facilitated diffusion through glucose transporters like GLUT 1 [54]. An increase in blood glucose levels increases the permeability of this BBB, leading to a rise in the influx of substances that might damage the brain [55]. Glucose molecules can interact with intra- and extracellular compounds, forming reactive oxygen species that can harm neurons and cells in the CNS. Changes in intelligence, learning, memory, and fine motor functions have been reported in children with T1D and explained by this mechanism [56]. Another molecule concentration controlled by insulin is ceramide, a precursor of many sphingolipids and an important component of the brain. Ceramide is a liposoluble molecule that can cross the BBB. The latest research shows that it might increase the risk of central insulin resistance, as it does for skeletal muscles [57]. This shows the importance of the peripheral action of insulin in keeping blood glucose levels low to protect brain metabolism.

The brain is also directly affected by insulin via the various insulin receptors located in it. Its specific and neuronal receptors cover a wide range of physiological actions, including brain metabolism, regulation of energy disposal, reproduction, dopamine signaling, glucose, and adipose tissue metabolism, and energy homeostasis [58]. These receptors are part of the modulation of satiety and the energy expenditure function of the brain. This makes insulin play a major role in this field, as it can work on the mesolimbic circuitry, changing eating and reward behaviors. It acts on the nucleus accumbens (Nac) and ventral tegmental area (VTA), causing satiety at high levels and increasing eating and obesity at low levels [58]. New clinical trials are demonstrating this by injecting patients with intranasal insulin, therefore bypassing the BBB and mimicking the action of central insulin. While not all trials present the expected results [59], intranasal insulin in the postprandial state was shown to reduce appetite, increase signaling in the amygdala, and, in some cases, increase the desire to eat only low-caloric food [59,60], making it a potential future target for obesity treatment and prevention.

The effect of insulin on neurodegeneration and the progression of certain neurologic disorders like Alzheimer’s disease (AD) is a new topic of discussion, as a strong link is being proven in the latest studies. Using functional magnetic resonance imaging and neuropsychology testing, the effects of insulin deprivation in T1D patients can be monitored. Comparing patients without diabetes and with T1D, differences in functional connectivity between regions, especially the cortical and hippocampus-caudate regions, were shown. In addition, participants with T1D showed lower baseline brain N-acetyl aspartate and myo-inositol levels but higher cortical fractional anisotropy, signifying unhealthy neurons and brain microscopy [61]. Other neurodegenerative disorders were also linked to disturbed insulin function or levels. *α*-synuclein, a known biomarker of neurodegeneration that deposits in cerebral neuronal cell bodies of people with Lewy body dementia, including parkinsonism, is present in elevated levels in children with T1D and obesity [62]. As mentioned before, one of the proposed mechanisms for neuro-inflammation is the production of reactive oxygen species and oxidative stress. This will disrupt the BBB integrity and microglia involvement, activating the inflammatory pathways. This neuro-inflammation is a known cause of cerebral neurodegeneration [62,63,64]. Kellar D et al. used intranasal insulin on AD patients and controls for 12 months. [65]. The trials showed an increase in interferon gamma (IFN-*γ*) and CSF eotaxins, which are associated with a slower cognitive decline in AD patients and a decrease in interleukin 6 (IL-6), which is associated with inflammatory states. A protective effect on the blood vessels was also shown with an increase in vascular endothelial growth factors (VEGF) and a decrease in serum amyloid A (SSA) associated with damaged blood vessels. Another clinical trial that also exposed AD patients to intranasal insulin for 12 months showed a decrease in white matter degradation, suggesting a protective effect on cognitive function [66].

Recent evidence suggested that prediabetes and diabetes are major risk factors for poorer cognitive performance [67,68]. In a recent trial, named the Finnish Geriatric Intervention Study to Prevent Cognitive Impairment and Disability (FINGER), the link between dysglycemia and cognitive function was examined. This study included 1259 at-risk individuals aged 60–77 years without dementia who were randomly assigned to a 2-year multidomain lifestyle intervention, which included nutrition advice, exercise, cognitive training, and social activities, or a regular health advice program. Participants underwent a cognitive function assessment at baseline, 12, and 24 months after enrollment using a modified Neuropsychological Test Battery (mNTB). Oral glucose tolerance tests (OGTTs) and brain MRI were conducted in parallel to link the glycemic state of patients (prediabetic, diabetic) with their mNTB outcomes. The results showed that higher baseline dysglycemia measures and insulin resistance were connected to less favorable changes in multiple cognitive measures and hippocampal volume [69]. Many trials confirmed the idea that intranasal insulin therapy improved the neurocognitive function of AD patients, as shown in a meta-analysis. The odds ratios and mean differences were calculated for the relevant results of 12 studies and favored the use of small doses of INI over placebos [70]. These results show intranasal insulin as a promising agent that can help delay the progression of neurocognitive diseases like Alzheimer’s disease.

The interaction between insulin and ApoE4 is a new topic of discussion in the AD studies. ApoE is a ligand for low-density lipoproteins (LDL), and the overexpression of its isoform ApoE4 is considered a risk factor for AD [71]. It works by increasing the cholesterol content of cell membranes and disrupting their normal function. This ligand is associated with an increase in neuroinflammation, which will disrupt the BBB and lead to oxidative stress-mediated neuronal damage as discussed before [72]. The overexpression of ApoE4 in neurons also results in increased tau phosphorylation, leading to an increase in insoluble cytoskeletal elements and neurofibrillary tangles. This overload of cholesterol in the cell membrane leads to the formation of amyloid β protein precursor internalization and amyloid formation, which raises the risk of angiopathy and intraparenchymal hemorrhage [71,73]. ApoE4 indirectly affects insulin function via the low-density lipoprotein receptor-related protein 1 (LRP1), which can directly interact with the beta subunit of the insulin receptor and regulate glucose uptake. ApoE4 inhibits LRP1, therefore playing a role in the development of insulin resistance in the context of AD [74,75,76], and the downregulation of the protective effects of insulin previously mentioned. These studies might explain why some intranasal insulin clinical trials failed to improve cognitive functions in some patients. They highlight the importance of considering other factors that directly or indirectly affect the action of central insulin.

### 3.5. The Role of Insulin in the Reproductive System

The reproductive system is also directly affected by insulin, particularly in the case of polycystic ovary syndrome (PCOS). Being the most prevalent endocrine disorder in females (10–15% of reproductive-age females), PCOS is linked with insulin resistance, hyperinsulinemia, and the metabolic complications of these conditions [77]. A large variety of markers for insulin resistance have been used over the years, each having its own advantages and limitations [78]. Both adipose tissue and skeletal muscles are directly affected by this insulin resistance, leading to abnormal glucose and fat metabolism. This drives an additional production of insulin by the pancreas, causing hyperinsulinemia that will worsen the resistance and start a vicious cycle [79]. Without enough suppression by insulin, lipolysis will increase free fatty acid (FFA) levels in the blood, further increasing insulin resistance and damaging the liver. This fat can be deposited in adipose tissues and contribute to this worsening cycle [80]. Excess FFAs and glucose shunted to the liver will cause inflammation of hepatic tissue, increase ALT levels, and increase the chances of liver steatosis and metabolic dysfunction-associated steatotic liver disease (MASLD) [77]. In fact, new trials are confirming that insulin resistance and obesity in PCOS women are risk factors that cause these women to be two times more likely to develop MASLD [81]. In addition, it is well understood that IR increases the risk of developing T2D and cardiovascular disease, making women with PCOS more prone to develop these diseases and their complications [82].

Hyperinsulinism is also linked to other PCOS symptoms and complications. Thecal steroidogenesis is directly affected by insulin via insulin receptors that stimulate the expression of CYP17*α*1 mRNA. Increased stimulation of these receptors will increase LH production and LH-mediated androgen production [83]. This increased androgen production that causes symptoms like acne, hirsutism, and irregular menstrual cycles was thought to be genetically driven. However, new research suggests that increased insulin levels play a major role in this mechanism, activating pituitary and adrenal gland cells along with ovarian theca cells to produce more androgens [84]. Additional pathologies that are associated with hyperinsulinism and insulin resistance are increased apoptosis of granulosa cells, increased risk of spontaneous abortions, and other pregnancy-related issues [85,86].

While PCOS presents with many complications, new clinical trials are showing that lifestyle changes like a healthier diet and exercise are effective in reducing the severity of these complications, especially the ones involving body metabolism and cardiovascular diseases [87,88]. Some women who presented with depression because of PCOS-driven hormonal imbalances showed improved emotional well-being after basic lifestyle changes [89]. Pharmacological treatment is also used in some cases. Metformin is one of the most prescribed drugs for women with PCOS because it can improve insulin sensitivity, decrease insulin levels, and decrease the chance of pregnancy complications [90,91]. Recent studies are exploring the benefits of other medications like liraglutide (GLP-1 receptor agonist) and N-acetylcysteine, which are showing promising results [92,93].

### 3.6. Cardiovascular Action of Insulin

Insulin also exerts significant effects on the cardiovascular system. Its receptors are extensively present on the surfaces of cells lining the vascular walls and the myocardium. When insulin binds to these receptors, it causes receptor phosphorylation and activation through its intrinsic kinase activity. This results in the phosphorylation of the insulin receptor and the activation of two major pathways: the phosphatidylinositol 3-kinase (PI3K) pathway and the mitogen-activated protein kinase (MAPK) pathway [94].

The PI3K pathway mediates the metabolic effects of insulin and activates endothelial nitric oxide synthase (eNOS) [95]. The nitric oxide (NO) generated by eNOS reduces vascular tone, inhibits proliferation of vascular smooth muscle cells, and decreases the adhesion of inflammatory cells and platelet aggregation at the endothelium. Additionally, insulin enhances eNOS phosphorylation in endothelial cells, boosting eNOS activity and substantially lowering reactive oxygen species production. Insulin also influences the synthesis of prostaglandins and other endothelium-derived factors that function as important vasodilators [96]. The MAPK pathway is responsible for mediating insulin’s influence on growth, cell division, and differentiation [97]. Furthermore, it facilitates various atherothrombotic effects, including increased production of plasminogen activator inhibitor-1 (PAI-1) and elevated expression of vascular cell adhesion molecules like VCAM-1. Also, a signaling cascade involving the MAPK pathway triggers the release of endothelin (ET-1), which can induce vasodilation via ET_B_ receptors on endothelial cells. However, ET-1 is more typically linked to vasoconstriction through ET_A_ receptors on vascular smooth muscle cells. Since insulin stimulates the release of both ET-1 and NO, the vasodilatory effects of insulin are often only observable when ET-1 activity is blocked with an antagonist [98].

The endothelium plays a crucial role in preserving vascular integrity. Vascular homeostasis maintains delivering oxygen and nutrients to tissues, conserves optimal vascular tone, regulates hemostasis, and modulates inflammation. However, in some conditions, the function of endothelial cells can be impaired, particularly when it occurs alongside the onset of insulin resistance. Remarkably, in individuals with obesity and diabetes, insulin’s vasoconstrictive and proatherogenic effects tend to dominate. A hallmark of insulin resistance is the selective impairment of the PI3K pathway, leading to decreased processes such as NO-mediated vasodilation, while insulin continues to activate MAPK-dependent pathways that contribute to atherothrombosis. Moreover, hyperinsulinemia and insulin resistance are commonly accompanied by increased LDL levels, which significantly promote the downregulation of eNOS expression, further impairing vascular function.

A study conducted in mice by Kubota et al. revealed that insulin signaling in endothelial cells is notably diminished following a high-fat diet, overlapping with a decrease in insulin-induced capillary recruitment and lower levels of insulin in the interstitial space [99]. Consequently, the endothelial insulin signaling pathway, essential for transporting insulin to the interstitial tissue, can be suppressed through dietary influences. Additionally, a recent investigation involving obese women with postprandial hyperglycemia employed microdialysis to show that higher circulating insulin levels were necessary to achieve interstitial insulin concentrations comparable to those in lean individuals. This altered insulin gradient was observed in both skeletal muscle and adipose tissue; notably, in healthy women, interstitial insulin in adipose tissue closely matched plasma insulin levels, whereas obese women exhibited a much greater disparity, indicating a more pronounced impairment [100]. Overall, these findings suggest that compromised insulin delivery in obesity may play a role in the development of metabolic insulin resistance and diabetes.

Hypertension is a frequent medical condition resulting from elevated peripheral vascular resistance and, in certain instances, heightened cardiac output [101]. It is characterized by increased baseline sympathetic nervous system activity and an overactivation of the renin–angiotensin–aldosterone system (RAAS). Individuals with hypertension often exhibit elevated insulin levels and impaired glucose tolerance [102]. A meta-analysis involving 10,230 participants with hypertension indicated that fasting insulin levels and insulin resistance (IR) are independent predictors of developing hypertension; specifically, the relative risk (RR) associated with fasting insulin was 1.54, with women showing a higher risk than men [103].

RAAS comprises hormones vital for maintaining stable blood pressure in the arteries. Renin converts angiotensinogen into angiotensin I (Ang I), which is then transformed into angiotensin II (Ang II) by angiotensin-converting enzyme (ACE). Ang II raises blood pressure by binding to AT1 and AT2 receptors, promoting vasoconstriction and sodium retention. Specifically, Ang II acts on the proximal tubules and the adrenal zona glomerulosa to stimulate aldosterone secretion, which further enhances sodium and water reabsorption in the distal tubules. Elevated RAAS activity in visceral fat (VAT) correlates with increased BMI. There is a positive relationship between RAAS activity and body weight, and RAAS activity tends to decrease following weight loss. In adipose tissue, RAAS contributes to and worsens insulin resistance by disrupting insulin-mediated glucose uptake and generating reactive oxygen species, which further impair insulin signaling [104]. Hyperinsulinemia can create a vicious cycle involving vascular damage, proliferation of smooth muscle cells, atherogenesis, calcium overload in cells, and increased renal sodium reabsorption [105]. Additionally, high insulin levels may synergistically activate the MAPK pathway alongside RAAS. In individuals with diabetes, RAAS is often upregulated, leading to elevated plasma renin, increased arterial pressure, and heightened renal vascular resistance [106].

Hyperglycemia promotes increased local production of Ang II and amplifies tissue responsiveness to it. Additionally, elevated blood glucose leads to the formation of advanced glycation end products (AGEs), which further stimulate the Ang II/AT1R pathway. It also results in the downregulation of ACE2, an enzyme responsible for generating Ang 1–7, thereby exacerbating the imbalance within RAAS [106]. Overall, the use of RAAS inhibitors can counteract the activation of RAAS caused by diabetes and mitigate its associated effects on hypertension and vascular damage.

Furthermore, insulin influences a wide array of functions within the heart, engaging multiple signaling pathways. The heart is an insulin-dependent organ, relying on insulin to facilitate glucose as its main energy substrate. Insulin signaling impacts various myocardium cells, including cardiomyocytes, fibroblasts, and endothelial cells. Its roles include reducing myocardial oxygen consumption, enhancing cardiac efficiency, aiding in myocardial relaxation, and increasing blood flow to the myocardium [107]. The signaling process involves insulin and IGF-1 receptors, which bind insulin, IGF-1/2, and insulin receptor substrates 1 and 2 (IRS1 and IRS2). These components are crucial regulators of cellular metabolism [108]. Additionally, pathways such as PI-3K, which activate Akt (also known as PKB), and the Janus kinase (JAK)2 pathway, play key roles in mediating the immediate effects of insulin on the cardiac muscle [108]. Therefore, insulin, by activation of the PI-3K pathway in cardiac muscle, leads to increased glucose uptake and metabolism via downstream effectors like Akt. This pathway supports the energetic and survival needs of cardiac cells, especially during increased workload or stress, ensuring proper cardiac function.

Insulin resistance observed in individuals with obesity can lead to left ventricular hypertrophy (LVH) through a complex interplay of factors. It contributes to increased hypertension, which is a major driver of LVH, and also directly affects cardiomyocytes, promoting their growth. Additionally, insulin resistance can cause metabolic changes and inflammation within the heart, further contributing to the development of LVH and its associated complications [109]. Also, it alerts cardiac metabolism by influencing glucose and fatty acids utilization, potentially impairing energy production and reducing cardiac efficiency, especially under stress.

### 3.7. The Link Between Insulin and the Kidney

Beyond the traditional crosstalk between insulin and its primary insulin-responsive organs, recent research indicates that insulin plays an important role in the kidneys. First reports regarding insulin’s effects on the kidney date back to the 1950s. Since then, the knowledge has expanded significantly; however, some uncertainties still remain.

Insulin can influence all types of renal cells, including mesangial cells, podocytes, and tubular epithelial cells. Both insulin receptors are broadly present throughout the kidney [110]. The interaction between insulin and renal tissues is a complex, dynamic process that affects various parts of the kidney, from the glomerulus to the renal tubules. This interaction regulates multiple functions such as glomerular filtration, glucose production, sodium excretion, glucose uptake, ion transport regulation, and the prevention of apoptosis [111,112].

Podocytes are the major constituent cells of the glomerulus, having intercellular junctions that form filtration barriers to help maintain normal renal function. Furthermore, emerging evidence indicates that the kidneys play a vital role in maintaining overall glucose balance through various mechanisms, including the reabsorption of glucose from the glomerular ultrafiltrate within renal epithelial cells, the uptake and utilization of glucose to satisfy the body’s energy requirements, and the process of gluconeogenesis from non-carbohydrate sources [113].

Podocytes express the elements of the insulin-signaling cascade, such as insulin receptor and both IRS1 and IRS2, with IRS2 having the highest prevalence [112]. In podocytes, insulin enhances glucose uptake not only through glucose transporter 4 (GLUT4) but also via GLUT1.

Renal gluconeogenesis is impaired in individuals with diabetes and/or insulin resistance [114]. Some studies indicate that renal epithelial cells increase their glucose uptake twofold in response to insulin by translocating GLUT1 and GLUT4 transporters to the plasma membrane, thereby amplifying insulin’s effects on renal gluconeogenesis and systemic blood glucose levels [115]. Additionally, hyperinsulinemia has been shown to suppress glucose production while enhancing glucose uptake in renal epithelial cells [116,117]. For example, a hyperinsulinemic clamp study observed a 61% reduction in renal glucose output and about a 72% decrease in renal glutamine-driven gluconeogenesis rates, notably higher than those observed in the liver (25%), following insulin treatment [118]. Insulin also influences the transport of gluconeogenic substrates within the kidney [119]. These findings underscore the importance and extent of renal glucose production, revealing that renal glucose release is more sensitive to insulin regulation compared to the liver. Furthermore, increased renal gluconeogenesis during the postabsorptive state may contribute to hyperglycemia in type 2 diabetes.

In the kidney, impairment of IRS1-dependent inhibition of gluconeogenesis in the proximal tubule contributes to hyperglycemia, whereas IRS2-dependent signaling remains intact [120]. In an animal study, the importance of insulin signaling in maintaining systemic glucose balance in IRS1/IRS2 double knockout mice was highlighted, emphasizing the dual regulation of gluconeogenesis by insulin signaling and glucose reabsorption. These findings align with earlier research indicating that individuals with diabetes in proximal tubules exhibit increased glucose reabsorption and insulin-dependent suppression of gluconeogenesis, which ultimately results in higher glucose production in the kidney than in the liver [121]. Given these observations, the regulation of renal gluconeogenesis remains a debated topic, as studies in rodent models with diabetes have reported both suppression and elevation of gluconeogenic gene expression [114,122]. These insights suggest new directions for exploring the role of insulin receptors in renal glucose homeostasis.

The capacity of kidney tubules to control sodium reabsorption is vital for maintaining blood volume and systemic blood pressure. Insulin influences various segments of the nephron to promote salt reabsorption. Specifically, in the proximal tubule, insulin enhances the activity of the Na+/H+ exchanger type 3 (NHE3) on the apical membrane and the sodium-bicarbonate cotransporter (NBCe1) on the basolateral membrane, leading to increased reabsorption of sodium and bicarbonate. The insulin-induced stimulation of NHE3 is probably mediated through the Akt pathway, which is known to facilitate the movement of NHE3 to the apical surface of proximal tubular cells via the PI3K pathway. Additionally, insulin promotes the activity of the Na-K-ATPase pump, further supporting sodium reabsorption in this segment [123]. In summary, insulin activates the mechanisms responsible for sodium uptake in the proximal tubule.

Research involving both lean and obese individuals showed that insulin-induced vasodilation through PI3K signaling is compromised in those with insulin resistance. Additionally, renal sodium reabsorption appears to be maintained or even increased in insulin-resistant states. In obese individuals with insulin resistance, insulin continued to reduce urinary sodium excretion, suggesting that insulin’s capacity to promote salt absorption remains intact. Collectively, these findings highlighted that enhanced insulin-stimulated salt reabsorption alongside impaired vasodilation play important roles in the development of hypertension associated with insulin resistance [124,125]. Table 1 presents the non-glycemic effects of insulin on selected systems.

## 4. Conclusions

Insulin plays a key role in maintaining metabolism, bone health, and muscle function while directly affecting conditions such as PCOS and neurodegeneration. A well-regulated activity will help preserve the good function of these systems, especially in patients prone to disrupted insulin activity, such as patients with T1D and T2D. Further research in this field is crucial to fully understand the action of insulin in each tissue and help develop medications that might not only treat but also prevent associated diseases.

## 5. Future Directions

The hypoglycemic action of insulin is well understood, its primary role is lowering blood glucose by promoting glucose uptake into cells and inhibiting glucose production by the liver. Consequently, this hormone is widely used for the management of hyperglycemia. Particularly in the case of T1D, insulin is a life-saving drug. The discovery of insulin in 1921 has been a landmark achievement in the history of medicine. However, this hormone also exerts many other extrapancreatic effects, which are still not fully understood or elaborated and require further exploration.

Further research into the multifaceted roles of insulin is essential to explore our understanding of its impact across various physiological systems. Studies should focus on elucidating the molecular mechanisms underlying insulin’s influence on skeletal remodeling, including its effects on osteoblast and osteoclast activity, to develop targeted therapies for osteoporosis and metabolic bone diseases. Animal models need to be developed with modified insulin signaling pathways specifically in bone tissue, complemented by longitudinal clinical studies to evaluate insulin’s impact on bone density, microarchitecture, and fracture risk. Especially, longitudinal studies on a large group of participants are needed in assessing the effects of various insulin analogues and insulin-sensitizing drugs on bone health, with implications for individuals with diabetes.

Exploring the interplay between insulin, IGF-1, and GH remains a promising direction, as these hormones form a complex regulatory network influencing growth, metabolism, and aging. Clarifying their interactions at cellular and systemic levels could lead to novel approaches for understanding growth disorders and metabolic syndromes.

Research into insulin’s role in protein and lipid metabolism is vital for understanding its contribution to energy homeostasis and obesity-related conditions. Investigations should aim to dissect how insulin modulates anabolic and catabolic pathways in various tissues, providing insights for managing insulin resistance and metabolic diseases. In particular, insulin resistance remains a significant medical issue among individuals with obesity. In the state of insulin resistance, signaling pathways are disrupted. Therefore, it is worth working on new medications that could help reduce insulin resistance. So far, we do not have a fully effective drug for treating insulin resistance. Metformin is one of the drugs that is widely used; however, it is often prescribed off-label for insulin resistance because it is not its primary indication. In addition, metformin is not always beneficial in treating insulin resistance.

Anti-obesity drugs, such as GLP-1 receptor agonists, seem promising in indirectly improving insulin resistance by reducing body mass in individuals with obesity. Excess body fat, particularly abdominal fat, is a major contributor to insulin resistance; therefore, GLP-1 receptor agonists might help increase the body’s sensitivity to insulin, potentially mitigating or even reversing the effects of insulin resistance. However, knowledge on this topic is still limited, and there is a need for multi-center studies to evaluate the effects of GLP-1 receptor agonists on improving insulin signaling across different body tissues in individuals with insulin resistance.

The effects of insulin on the CNS warrant further exploration, particularly regarding its influence on appetite regulation, cognition, and neurodegeneration. Understanding insulin signaling pathways in the brain could open new therapeutic avenues for conditions like Alzheimer’s disease and obesity.

Additionally, the role of insulin in the reproductive system, including ovarian function, should be investigated in order to understand fertility issues and reproductive health in metabolic disorders.

Finally, comprehensive studies are needed to elucidate insulin’s effects on the cardiovascular system and kidneys, especially in the context of diabetes-related complications. Such research could inform strategies to prevent or mitigate cardiovascular disease and renal dysfunction associated with insulin resistance and hyperglycemia.

To sum up, by broadening understanding of insulin’s diverse roles, there might be an opportunity to discover novel therapeutic targets and improve management of complex metabolic and non-metabolic diseases.

## Figures and Tables

**Figure 1 jcm-14-05039-f001:**
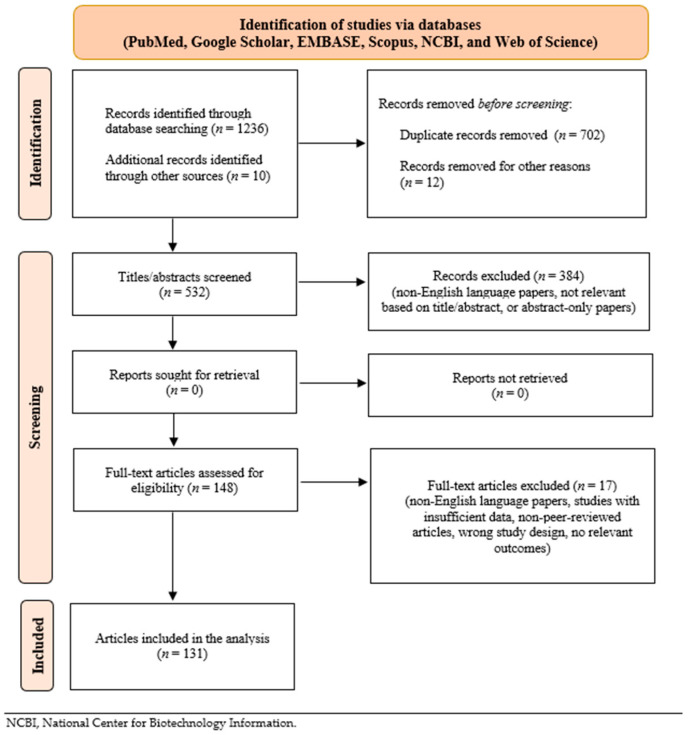
Flow chart of the review process.

**Figure 2 jcm-14-05039-f002:**
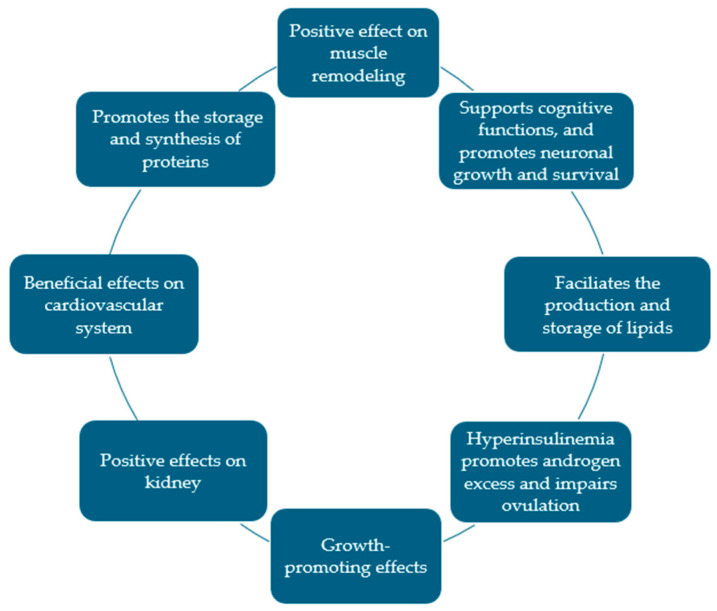
Summarization of the main insulin actions beyond controlling glucose homeostasis.

**Table 1 jcm-14-05039-t001:** Overview of the non-glycemic effects of insulin on selected systems [18,24,27,69,79,94,95,96,97,98,99,100,101,110,111,112,113,114,115].

Affected System	The Mechanism by Which Insulin Acts
SKELETAL SYSTEM	- Stimulates osteocalcin gene transcription and post-translational modification - Promotes osteoblast function, collagen synthesis, ALP production, and Runx2 expression - Modulates bone remodeling through IRS and RANKL regulation
GROWTH HORMONE	- Shares signaling pathways with IGF-1 to regulate growth, metabolism, and gene expression - Inhibits IGFBP-1, increasing IGF-1 receptor interaction and promoting growth
PROTEIN METABOLISM	- Suppresses whole body and muscle proteolysis - Enhances amino acid transport - Increases muscle blood flow (via NO) - Activates PI3K-Akt-mTORC1 to promote mRNA translation and protein synthesis - Reduces the risk of the development of sarcopenia
LIPID METABOLISM	- Suppresses lipolysis and reduces FFA flux - Enhances triglyceride, cholesterol, and glycogen synthesis postprandially in the liver - Coordinates systemic metabolic homeostasis
CENTRAL NERVOUS SYSTEM	- Protection of the brain metabolism by controlling glucose influx and ceramide levels in the CNS - Regulation of reproduction, dopamine signaling, glucose, and adipose tissue metabolism - Regulation of satiety and energy expenditure - Slowing cognitive decline by increasing interferon gamma (IFN-γ) and CSF eotaxins, and by decreasing IL-6 levels - Decreased white matter degradation
REPRODUCTIVE SYSTEM	- Increases LH levels and androgen production
CARDIOVASCULAR SYSTEM	- Activation of the PI3K pathway, decreases vascular tone, inhibits proliferation of vascular smooth muscles, decreases the adhesion of inflammatory cells and platelet aggregation - Lowers reactive oxygen species - Increases production of plasminogen activator inhibitor-1 and VCAM - Reduces myocardial relaxation, increases blood flow, glucose uptake and metabolism in the myocardium - Promotes growth of cardiomyocytes - Insulin resistance contributes to the development of left ventricular hypertrophy
RENAL SYSTEM	- Regulates renal gluconeogenesis - Enhances glucose uptake in renal podocytes - Enhances the activity of the Na+/H+ exchanger and sodium bicarbonate cotransporter, leading to an increased reabsorption of Na and HCO3- - Promotes the activity of the Na+/K+ ATPase, increasing Na+ reabsorption

ALP, alkaline phosphatase; IGF-1, insulin-like growth factor 1; IGFBP-1, insulin-like growth factor binding protein 1; NO, nitric oxide; FFA, free fatty acids; CNS, central nervous system; CSF, cerebrospinal fluid; IL-6, interleukin 6; LH, luteinizing hormone, VCAM, vascular cell adhesion molecule.

## Data Availability

Not applicable.

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
