# Peer review of "Understanding Insulin Actions Beyond Glycemic Control: A Narrative Review"

_jcm, 2025, doi:10.3390/jcm14145039_

Round 1

Reviewer 1 Report

Comments and Suggestions for Authors
  1. The review effectively addresses insulin’s multifaceted roles beyond glycemic control, including its effects on lipid/protein metabolism, muscle remodeling, neuroprotection, reproduction, and growth. The topic is timely and clinically significant.
  2. The manuscript is generally well written, logically structured, and accessible to readers. Clinical implications, especially regarding PCOS and neurodegeneration, are appropriately emphasized.
  3. Please clarify the literature selection process. Include details on databases searched, time frame, inclusion/exclusion criteria, and whether the search was systematic or selective.
  4. A more thorough evaluation of the strength and quality of cited studies (e.g., clinical vs. preclinical) would improve the scientific rigor of the review.
  5. Consider adding a summary figure or table outlining insulin’s systemic effects and related disorders to enhance clarity and visual appeal.
  6. Revise the incomplete sentence: “...especially in patients prone to disrupted insulin activity, such as patients with...” to ensure clarity and coherence.
  7. Support claims particularly those on bone health and neurodegeneration with specific examples or citations.
  8. Include a brief discussion on research gaps and future directions, such as targeting insulin signaling in non-glycemic contexts.

Author Response

Response to Reviewer 1

Thank you for the time and effort in reviewing and providing feedback on our manuscript, and we are grateful for the insightful comments and valuable improvements to our paper. Below, we provide the point-by-point responses.

The review effectively addresses insulin’s multifaceted roles beyond glycemic control, including its effects on lipid/protein metabolism, muscle remodeling, neuroprotection, reproduction, and growth. The topic is timely and clinically significant. The manuscript is generally well written, logically structured, and accessible to readers. Clinical implications, especially regarding PCOS and neurodegeneration, are appropriately emphasized.

  1. Please clarify the literature selection process. Include details on databases searched, time frame, inclusion/exclusion criteria, and whether the search was systematic or selective.

Answer: Thank you for the valuable comment. We improved the methodology section by clarifying inclusion/exclusion criteria and explaining the measures to mitigate selection bias. Despatie is not a systematic review; we also prepared a PRISMA-style flow diagram detailing the search process. Please see the Materials and Methods section and Figure 1.

  1. A more thorough evaluation of the strength and quality of cited studies (e.g., clinical vs. preclinical) would improve the scientific rigor of the review.

 Answer: Thank you for that suggestion. As suggested, we improved the strength and quality by adding more details regarding cited studies.

  1. Consider adding a summary figure or table outlining insulin’s systemic effects and related disorders to enhance clarity and visual appeal.

Answer: Thank you for that comment. In our previous version of our manuscript, Table 1, which gives an overview of the non-glycemic effects of insulin on selected systems, was included; therefore, the purpose of this table was to summarize the pathways of insulin action. No information was added about diseases that can result from insulin dysfunction in extrapancreatic systems. We had the impression that the table would be overloaded, which would hinder understanding of the main goal. We hope you understand our point of view.

  1. Revise the incomplete sentence: “...especially in patients prone to disrupted insulin activity, such as patients with...” to ensure clarity and coherence.

Answer: Done

  1. Support claims particularly those on bone health and neurodegeneration with specific examples or citations.

Answer: Done

  1. Include a brief discussion on research gaps and future directions, such as targeting insulin signaling in non-glycemic contexts.

Answer: Thank you for that valuable comment. As suggested, we added a section discussing this issue.

Reviewer 2 Report

Comments and Suggestions for Authors

Congratulations for your work. In a time when there are gathering a lot of data regarding diabetes, metabolic disease, intermuscular adiposity as markers of inflamation, it is very useful to see a paper dealling with different effect of insulin. I think you have to study deeply it’s effect on muscle development and central nervous system. 

Author Response

Response to Reviewer 2

We would like to thank the reviewer for the careful and thorough reading of this manuscript and their critical assessment of our work. We have taken the comments on board to improve and clarify the manuscript. In the following, we address their concerns point by point.

Major comments:

  1. Congratulations for your work. In a time when there are gathering a lot of data regarding diabetes, metabolic disease, intermuscular adiposity as markers of inflamation, it is very useful to see a paper dealling with different effect of insulin. I think you have to study deeply it’s effect on muscle development and central nervous system.

Answer: Thank you for that suggestion. We added some information as suggested.

Reviewer 3 Report

Comments and Suggestions for Authors

The manuscript Is very interesting but Is very incomplete.In fact, It completely lacks to describe the action of insulin on the cardiovascular system and the kidney. It may also be useful to explain what happens at the level of different tissues in the presence of hyperinsulinemia associated with insulin resistance. Therefore, the manuscript can not be accepted in its current form, unless completed with these informations.

Author Response

Response to Reviewer 3

We would like to thank the reviewer for the careful and thorough reading of this manuscript and their critical assessment of our work. We have taken the comments on board to improve and clarify the manuscript. In the following, we address their concerns point by point. The changes to the manuscript are shown in an underlined font.

Major comments:

  1. The manuscript Is very interesting but Is very incomplete.In fact, It completely lacks to describe the action of insulin on the cardiovascular system and the kidney. It may also be useful to explain what happens at the level of different tissues in the presence of hyperinsulinemia associated with insulin resistance. Therefore, the manuscript can not be accepted in its current form, unless completed with these informations.

Answer: Thank you for that valuable comment. As suggested, we included in the manuscript the discussion on the action of insulin on the cardiovascular system and the kidney. Also, we mentioned that some information happens at the level of different tissues in the presence of hyperinsulinemia associated with insulin resistance.

Round 2

Reviewer 3 Report

Comments and Suggestions for Authors

The authors have revised the manuscript according to the Reviewers' suggestions further improving the manuscript.